# Sex-Dependent Effects of Chronic Restraint Stress on Mood-Related Behaviours and Neurochemistry in Mice

**DOI:** 10.3390/ijms241210353

**Published:** 2023-06-19

**Authors:** Mauritz Frederick Herselman, Liying Lin, Shayan Luo, Akihiro Yamanaka, Xin-Fu Zhou, Larisa Bobrovskaya

**Affiliations:** 1Health and Biomedical Innovation, Clinical and Health Sciences, University of South Australia, Adelaide, SA 5000, Australia; mauritz.herselman@mymail.unisa.edu.au (M.F.H.); liying.lin@mymail.unisa.edu.au (L.L.); zhou0010@gmail.com (X.-F.Z.); 2Department of Breast Surgery, Xiangya Hospital, Central South University, Changsha 410008, China; 3Chinese Institute for Brain Research, Beijing 102206, China; yamank@cibr.ac.cn

**Keywords:** stress, anxiety, hippocampus, adrenal, monoamine pathways, restraint

## Abstract

Anxiety and depressive disorders are closely associated; however, the pathophysiology of these disorders remains poorly understood. Further exploration of the mechanisms involved in anxiety and depression such as the stress response may provide new knowledge that will contribute to our understanding of these disorders. Fifty-eight 8–12-week-old C57BL6 mice were separated into experimental groups by sex as follows: male controls (*n* = 14), male restraint stress (*n* = 14), female controls (*n* = 15) and female restraint stress (*n* = 15). These mice were taken through a 4-week randomised chronic restraint stress protocol, and their behaviour, as well as tryptophan metabolism and synaptic proteins, were measured in the prefrontal cortex and hippocampus. Adrenal catecholamine regulation was also measured. The female mice showed greater anxiety-like behaviour than their male counterparts. Tryptophan metabolism was unaffected by stress, but some basal sex characteristics were noted. Synaptic proteins were reduced in the hippocampus in stressed females but increased in the prefrontal cortex of all female mice. These changes were not found in any males. Finally, the stressed female mice showed increased catecholamine biosynthesis capability, but this effect was not found in males. Future studies in animal models should consider these sex differences when evaluating mechanisms related to chronic stress and depression.

## 1. Introduction

Anxiety and depressive disorders are closely associated and have a substantial socioeconomic burden worldwide; however, the pathophysiology of these disorders remains poorly understood [1,2,3,4]. The incidence rates of depression are higher in females than in males, yet the underlying factors contributing to this phenomenon remain to be elucidated [1]. In the last few decades, evidence in preclinical and clinical studies has provided strong support for the involvement of the depletion of the monoamine neurotransmitters dopamine, noradrenaline and serotonin in anxiety and depression; however, in recent years it has become apparent that therapies which target these alterations have delayed effects and are ineffective in 50% of previously untreated individuals [1,5]. It has also now been controversially suggested that there is insufficient consistent evidence to support decreased serotonin levels in depression [6]. Yet, recent studies still consistently show that the activity of monoamine oxidase (MAO), responsible for the degradation of monoamines, is increased in depression [1]. Recent evidence has also implicated the kynurenine pathway and the gut–brain axis in depression, given it modulates the availability of the serotonin precursor, tryptophan, placing the role of monoamines in anxiety and depression back into the “spotlight” [7,8]. Nonetheless, further exploration of other mechanisms involved in anxiety and depression such as the stress response may still provide new knowledge that may contribute to our understanding of these disorders.

Exposure to stress leads to the adrenal release of glucocorticoids such as cortisol through hypothalamic–pituitary–adrenal (HPA) axis activation, while the adrenal release of the catecholamines adrenaline and noradrenaline is mediated by the sympathoadrenal medullary (SAM) system [9]. Glucocorticoids not only regulate the HPA axis via negative feedback, but also modulate neuronal excitability via the glutamatergic modulation of synaptic plasticity, particularly in highly neuroplastic brain regions such as the hippocampus and the prefrontal cortex (PFC) [10]. This glucocorticoid-mediated regulation of synaptic plasticity shares multiple interactions with mature brain-derived neurotrophic factor (mBDNF), a neurotrophin which further refines neuroplastic processes, closely associating neuroplasticity with the HPA axis [10]. Chronic stress is known to impair glucocorticoid receptor binding, consequently leading to impairments in the negative feedback of the HPA axis and neuroplasticity [11]. Rodent models of chronic stress have also shown a reduction in the expression of the PFC and hippocampal SNARE proteins SNAP-25 and VAMP-2, which may lead to a reduction in neurotransmission of monoamine neurotransmitters as well as glutamate [12]. Meanwhile, peripheral adrenaline, which cannot cross the blood–brain barrier, is known to invoke emotional responses in the brain indirectly by by-passing the blood–brain barrier through the activation of α- and β-adrenergic receptors on the vagus nerve [13]. Vagal nerve activation in this way directly stimulates noradrenaline release from the locus coeruleus which activates the amygdala via adrenergic receptors [13]. The amygdala then modulates memory consolidation in the hippocampus, but this is dependent on the binding of glucocorticoid receptors by glucocorticoids, in both the amygdala and the hippocampus [13]. Thus, both the HPA axis and the SAM system are closely associated in emotional-mediated behaviours. Furthermore, while it is well established that the HPA axis is dysregulated in depression [9,14], the SAM system has not received the same level of attention in the context of the disorder.

The endpoint of the SAM system, namely, the adrenal catecholamine biosynthesis pathway, is well characterised. Beginning with the stimulation of adrenal medullary chromaffin cells via the splanchnic nerve, catecholamine biosynthesis is rate-limited by tyrosine hydroxylase (TH) which converts tyrosine to ʟ-DOPA [15,16]. Downstream from this reaction, dopamine is converted into noradrenaline by dopamine β–hydroxylase followed by the conversion of noradrenaline to adrenaline by phenylethanolamine N-methyltransferase (PNMT) [15]. Thus, the regulation of the adrenal TH activity governs the subsequent catecholamine output from the adrenal glands; however, its regulation is complex since it is influenced by almost every form of regulation including phosphorylation [16,17]. According to Dunkley and Dickson [16], this phosphorylation of TH occurs at the serine residues (Ser) 19, 31 and 40, of which the latter two are directly involved in TH activation. Most studies of rodent models of chronic stress have only evaluated the protein levels or mRNA levels of adrenal catecholamine-synthesising enzymes, reporting inconsistent results likely due to the heterogeneity of protocols across the studies [18,19,20]. Moreover, these studies only used male mice; however, Manni et al. [21] previously showed potential sex-specific mechanisms, with adrenal TH protein levels being increased in the presence of chronic restraint stress in female mice, but not in males. It is yet to be established whether or not sex differences exist in the regulation of adrenal TH by phosphorylation events in a situation of chronic stress.

Thus, the present study aimed to firstly investigate the effects of randomised chronic restraint stress (RS) on the regulation of tryptophan metabolism and neuroplasticity in key brain regions involved in neuropsychiatric disorders in male and female C57BL6 mice. Second, we aimed to investigate the effects of chronic RS on the regulation of adrenal catecholamine biosynthetic enzymes in both sexes to provide further understanding of the regulation of TH in chronic stress.

## 2. Results

### 2.1. Behavioural Tests

#### 2.1.1. Open Field Test

The locomotive and anxiety-like behaviours were assessed using the open field test (OFT) and the elevated plus-maze (EPM) test. In the OFT (Figure 1), a main effect of sex ($F_{1,52}$ = 3.47; *p* = 0.0682) and a significant main effect of RS ($F_{1,52}$ = 22.01; *p* < 0.0001) were found for the total distance travelled in the test (Figure 1A), with a significant increase in the total distance travelled for male RS mice compared with male controls (4.328 ± 1.12 m, *p* = 0.0006). For females, a similar significant increase in the distance travelled was found for the RS group compared with female controls (3.095 ± 1.12 m, *p* = 0.0158). These results suggested that RS increased the locomotive behaviour in the OFT regardless of sex, although this effect was more pronounced in males.

A significant main effect of sex ($F_{1,52}$ = 13.30; *p* = 0.0006) was found for the time spent in the central zone (Figure 1B) as a measure of anxiety-like behaviour. No differences were found between the control and the RS groups; however, the female mice spent more time in the central zone overall and significantly more time in this zone compared with the male RS group (34.050 ± 9.50 s, *p* = 0.0015), suggesting that males may demonstrate higher levels of anxiety-like behaviour in the OFT.

Since significant differences were found for locomotive behaviour, the number of entries to the central (Figure 1C) and peripheral zones (Figure 1D) were corrected per the total distance prior to the analysis. A significant main effect of sex ($F_{1,52}$ = 56.12; *p* < 0.0001) was found for the number of entries to the central zone, with the female mice showing a higher number of entries to the central zone compared to the males in both the control (0.927 ± 0.21, *p* = 0.0001) and the RS (1.371 ± 0.22, *p* < 0.0001) groups. The male RS mice also showed a reduction in the number of entries compared to the male controls (*p* = 0.0628). These results further corroborated those found for the time spent in the central zone. A significant main effect of sex ($F_{1,52}$ = 56.99; *p* < 0.0001) was also found for the number of entries to the peripheral zone, with similar increases in entries to the zone for the female mice compared to the males in both the control (0.923 ± 0.21, *p* = 0.0001) and the RS (1.367 ± 0.22, *p* < 0.0001) groups. A reduction in the number of entries to the peripheral zone was found for the male RS group compared with the male controls (*p* = 0.0584), suggesting that the male mice in the RS group demonstrated higher levels of anxiety-like behaviour compared with their control counterparts.

#### 2.1.2. Elevated Plus-Maze Test

In the EPM (Figure 2), the locomotive behaviour was also assessed using the total distance measure (Figure 2A). A significant main effect of RS ($F_{1,52}$ = 4.47; *p* = 0.0393) was found. In males, no differences were evident between the control and the RS groups; however, for females, the RS mice increased their locomotive behaviour relative to the female controls (2.439 ± 0.85 m, *p* = 0.0124). These results suggest that RS only increased the locomotive behaviour in female mice in the EPM.

The number of open arm entries (Figure 2B) was then assessed, as mice with high levels of anxiety-like behaviour will typically avoid these open areas. A significant main effect of sex ($F_{1,52}$ = 10.13; *p* = 0.0025) was found, with females showing fewer entries to the open arms overall; however, a significant reduction in open arm entries was only found between the male and the female control groups (−3.300 ± 1.10, *p* = 0.0085). Similarly, for the number of entries to the closed arms (Figure 2C), a significant main effect of RS ($F_{1,52}$ = 6.73; *p* = 0.0123) was found. The female control mice showed significantly fewer entries to the closed arms compared with the male controls (−3.662 ± 1.59, *p* = 0.0504). The female RS mice showed a significant increase in the number of closed arm entries compared to the female controls (4.579 ± 1.62, *p* = 0.0133), suggestive of an increased locomotive behaviour.

As an additional measure, the time spent in the open (Figure 2D) and closed arms (Figure 2E) was measured. A significant main effect of sex ($F_{1,52}$ = 5.85; *p* = 0.0191) was found for the time spent in the open arms, with female mice generally spending less time in these arms; however, a significant difference was only found between female and male controls (−13.16 ± 4.97 s, *p* = 0.0212). No differences were found between any of the groups for the time spent in the closed arms. Together, these results suggest that in the EPM, the female mice exposed to RS showed higher levels of locomotive behaviour compared to their control counterparts, while the male mice appeared to show no differences in behaviour when comparing the control and the RS groups.

#### 2.1.3. Sucrose Preference Test and Tail Suspension Test

Next, depression-like behaviours related to anhedonia and behavioural despair were evaluated using the sucrose preference test (SPT) and the tail suspension test (TST), respectively. In the SPT (Figure 3A), a significant main effect of RS ($F_{1,44}$ = 4.10; *p* = 0.0490) was found, with no further differences, suggesting that RS reduced sucrose consumption overall, regardless of sex. In the TST (Figure 3B), no significant differences were found as a result of RS; however, a significant main effect of sex ($F_{1,51}$ = 5.40; *p* = 0.0241) was found, with females showing less time spent immobile, although this did not reach significance. These results suggest that RS increased the anhedonia-related depression-like behaviour, while it had no effect in the TST, which is a measurement of behavioural despair.

### 2.2. Serotonergic and Noradrenergic Regulation in the Prefrontal Cortex

The regulation of serotonin and noradrenaline pathways and transporters was investigated in the PFC (Figure 4). Two-way ANOVA of the protein levels of TPH2 showed a significant main effect of sex ($F_{1,22}$ = 13.21; *p* = 0.0015), with female controls showing higher levels than male controls (0.190 ± 0.05, *p* = 0.0040); however, no effects of RS were found. No differences were evident in the protein levels of IDO in the PFC. No significant differences were found for the MAO-A levels. The serotonin transporter (SERT) and noradrenaline transporter (NET) protein levels were also unaffected by RS and showed no differences between the sexes. The analysis of the VMAT2 protein levels showed a significant main effect of RS ($F_{1,22}$ = 10.18; *p* = 0.0042), with the female RS mice showing a reduction in the VMAT2 levels compared with female controls (−0.148 ± 0.04, *p* = 0.0046). These results suggest that female mice exposed to RS may have had a reduction in presynaptic monoamine transport, while male mice were more resilient to these effects.

### 2.3. mBDNF and Synaptic Protein Expression in the Prefrontal Cortex

Next, we investigated the levels of mBDNF and synaptic proteins in the PFC (Figure 5). The analysis of the mBDNF protein levels revealed a significant main effect of sex ($F_{1,22}$ = 5.521; *p* = 0.0282), with mBDNF levels being higher in female RS mice than in male RS mice (*p* = 0.0532). The analysis of SNAP25 revealed a significant main effect of sex ($F_{1,22}$ = 27.83; *p* < 0.0001), with female mice showing higher protein levels compared with males in both the control (0.661 ± 0.21, *p* = 0.0098) and the RS (0.858 ± 0.20, *p* = 0.0005) groups. However, no effects of RS were found.

The analysis of the VAMP2 protein levels showed a significant main effect of sex ($F_{1,22}$ = 38.58; *p* < 0.0001) as well as a significant interaction effect ($F_{1,22}$ = 4.51; *p* = 0.0453). the female mice once again showed higher levels of VAMP2 compared with the males in both the control (0.404 ± 0.07, *p* < 0.0001) and the RS (0.1983 ± 0.07, *p* = 0.0129) groups. PSD95 was also analysed; however, no differences were found between any of the groups.

### 2.4. Serotonergic and Noradrenergic Regulation in the Hippocampus

Similarly to the PFC, proteins involved in the regulation of serotonin and noradrenaline were investigated in the hippocampus (Figure 6). The protein levels of TPH2, IDO and MAO-A were unaffected by RS, and no effects of sex on their expression levels were observed. The analysis of the hippocampal SERT protein levels showed a significant main effect of sex ($F_{1,22}$ = 6.98; *p* = 0.0149), with the female RS mice showing lower levels compared with the male RS mice (−0.6420 ± 0.22, *p* = 0.0159). The VMAT2 protein levels were unaffected by both RS and sex. The analysis of the NET levels showed a significant main effect of sex ($F_{1,22}$ = 11.66; *p* = 0.0025), with the female controls showing a significant reduction compared with the male controls (−0.2671 ± 0.10, *p* = 0.0206).

### 2.5. mBDNF and Synaptic Protein Expression in the Hippocampus

Synaptic proteins in the hippocampus were also investigated (Figure 7). A significant main effect of sex ($F_{1,22}$ = 9.99; *p* = 0.0045) was found for the mBDNF protein levels, with a significant reduction in female RS mice compared with male RS mice (−0.317 ± 0.11, *p* = 0.0135). The analysis of the SNAP25 protein levels showed a significant main effect of RS ($F_{1,22}$ = 6.06; *p* = 0.0221) and a significant main effect of sex ($F_{1,22}$ = 16.71; *p* = 0.0005). A significant reduction in the SNAP25 levels was found in the female RS group compared with the female control group (−0.203 ± 0.07, *p* = 0.0137). Furthermore, the SNAP25 protein levels in the female RS mice were significantly lower than those in the male RS group (−0.281 ± 0.07, *p* = 0.0009), while the levels remained unchanged among the male mice, suggesting a sex-specific effect of RS on the hippocampal SNAP25 levels. The protein levels of hippocampal VAMP2 and PSD95 were unaffected by sex differences; however, a main effect of RS increasing the PSD95 protein levels overall was found ($F_{1,22}$ = 7.46; *p* = 0.0122).

### 2.6. Regulation of Adrenal Catecholamine Synthesis

Finally, the catecholamine synthetic capacity of the adrenal glands was evaluated by measuring the protein levels of TH and PNMT as well as the phosphorylation of TH (Figure 8). The phosphorylation of TH at Ser19 showed no differences between the groups, while pSer31TH showed a significant interaction effect ($F_{1,23}$ = 8.45; *p* = 0.0079), with the female RS mice having higher levels of phosphorylation compared with the male RS mice (0.321 ± 0.12, *p* = 0.0252). The analysis of pSer40TH revealed a significant interaction effect ($F_{1,22}$ = 8.76; *p* = 0.0070), with the female controls having lower levels of phosphorylation compared with the male controls (*p* = 0.0761). The male RS mice showed a decreased trend in pSer40TH compared with the male controls, while the female RS mice showed an increase compared with their control counterparts (*p* = 0.0896). These results suggest that the phosphorylation of TH, at least at Ser31 and Ser40, may be differentially regulated by sex.

The protein levels of TH were not affected by sex or by RS. The analysis of the PNMT protein levels revealed a significant main effect of sex ($F_{1,22}$ = 5.62; *p* = 0.0265), with females tending to have higher levels overall, but the female RS mice had higher PNMT protein levels compared with the male RS mice, although this difference did not reach significance (*p* = 0.0621).

## 3. Discussion

The present study aimed to investigate potential sex-dependent effects of chronic RS on behaviour, neurochemistry and adrenal catecholamine synthesis regulation. Behavioural testing using the OFT showed that 4 weeks of chronic restraint stress did not affect the thigmotactic behaviour in females; however, males spent less time in the central zone overall compared with females. We also found that male mice tended to show less exploratory behaviour in the open field overall, since they performed fewer crossings into each of the zones compared with females. According to O’Leary et al. [22], measures of the total distance travelled and zone entries in the OFT are considered a measure of exploratory behaviour. Chronic RS may have further reduced the exploratory behaviour in males, since a further reduction in zone entries was found in the male RS group compared to their control counterparts. Similar studies of male C57BL6 mice in chronic RS showed reductions in the time spent in the central zone in the OFT and in the open arms of the EPM; however, this may have been due to longer test durations in these studies [23] and differences in the RS protocols [23,24]. In the EPM in the present study, the locomotive behaviour was increased in females exposed to RS, and this finding was supported by a higher number of closed-arm entries and a greater distance travelled compared with controls. While studies of female rodent behaviour are limited, Simpson et al. [25] showed that female rats have greater baseline locomotion in the OFT and greater baseline velocity in the EPM; however, they found no differences in the time spent in the central zone or in the time spent in the open arms in each test, respectively, compared to males. Here, we found that stressed female mice exhibited increased locomotive behaviour in the EPM, but this effect did not occur in stressed male mice.

The assessment of depression-like behaviour showed that chronic RS produced anhedonic-like (inability to experience pleasure) behaviour regardless of sex in the SPT, but failed to produce behavioural despair in the TST, which is a measure of “learned helplessness” or a passive coping strategy during stress. Previous studies reported conflicting results regarding the effects of chronic stress on sucrose preference in male versus female rodents, with some studies reporting no effects of sex [26,27], and others reporting greater susceptibility in males to showing decreased sucrose preference [28,29] or greater susceptibility in females [30]. These conflicting findings may be due to differences in the concentration of sucrose used or other aspects of methodology; however, of note is a recent study that found food and water provision or deprivation in the SPT did not influence preference, with male and female Wistar rats performing similarly [31]. The discrepancies in the literature may also be due to the influence of the inbred strain of rodent used. Colelli et al. [32] showed that male C57Bl6J mice and DBA/2J mice engage separate memory systems during stress-associated memory consolidation, with C57Bl6J mice showing greater amygdala–hippocampal connectivity, and DBA/2J mice showing greater amygdala–dorsolateral striatal connectivity. Thus, different strains may respond to chronic restraint stress protocols differently by engaging different avenues of stress-associated memory consolidation.

In the present study, the female mice spent less time immobile overall in the TST, suggesting that female C57BL6 may respond differently from males in the TST. A 2-week chronic RS protocol (2 h per day for 14 consecutive days) using female mice was shown to have no effect on immobility in the TST [33], while a study that employed a chronic unpredictable mild stress protocol showed increased immobility in both males and females, although no sex-specific effects were identified [30]. Thus, the severity, and possibly the level of predictability, of the stressors employed may affect the outcomes in the TST. Altogether, the behavioural results showed that male and female mice of the same age and species produce different behaviours across the same behavioural tests. This may be due to several reasons such as the effects of the oestrous cycle in females potentially influencing the behavioural outcomes in anxiety-like and depression-like behavioural tests, and rodent behavioural tests being mostly validated in males and not females [25,34], but evidence is also building for sex differences in neural circuits, since some circuits mediate behaviour in each sex in the same manner, while other circuits can be sex-specific or have convergent effects on behaviour, highlighting the importance of considering sex as a factor in chronic stress and depression research [34].

The tryptophan and serotonin metabolism pathways and transporters were also investigated in the PFC and hippocampus. Previous studies of chronic RS found decreased TPH2 and increased IDO and MAO-A in male rodents [23]; however, sex differences have not yet been investigated in the context of chronic stress, to our knowledge. This is of importance, given female rodents are known to have higher levels of tryptophan and serotonin in both the brain and the circulation compared with males [35]. Indeed, the present study found higher levels of TPH2 in the PFC in female controls, suggesting higher basal levels of serotonin; however, TPH2 in the hippocampus as well as IDO and MAO-A in both the PFC and the hippocampus were unaffected by sex or chronic RS. The SERT protein levels were also unaffected by sex or chronic stress in the PFC; however, the SERT levels in the hippocampus were remarkably lowest in females exposed to RS, even though studies have shown hippocampal SERT is increased, at least in males, during chronic stress [36,37]. The hippocampal NET protein levels were also lower in female mice, but this is likely a basal characteristic, as the NET levels were unaffected by stress, even in males. Gavrilović et al. [38] showed that 2 weeks of chronic RS reduced hippocampal NET in male rats in association with increased VMAT2 levels, whereas our 4-week protocol did not have this effect; thus, the duration of the RS protocol may influence the NET levels. Another explanation may be that changes in the VMAT2 protein levels may be unrelated to NET, since other than monoamines, VMAT2 is also known to transport glycine, GABA and acetylcholine neurotransmitters, albeit to a lesser extent than monoamines [39]. This may also explain why the VMAT2 levels in the PFC were significantly reduced in stressed females, despite any changes in monoamine transporter levels, warranting further studies on the functional implications of PFC VMAT2 reductions in female rodents in chronic stress.

Differential effects were found in the levels of mBDNF and synaptic proteins across the PFC and the hippocampus. In the PFC, chronic RS did not have any effects on the protein levels of mBDNF, SNAP-25, VAMP2 or PSD-95. Müller et al. [40] previously showed 3 weeks of chronic RS decreased SNAP-25 and increased VAMP2 in the PFC of male rats; thus, in our study the longer duration of the RS protocol may have normalised the levels of these proteins as a compensatory mechanism, or it is also possible that the regulation of these proteins is species-specific. The mBDNF levels were higher in females than in males, which may have been due to the involvement of sex hormones, since an oestrogen response element has been identified in the BDNF gene, and the oestrous cycle dynamically regulates the phosphorylation of TrkB, a major BDNF receptor [41]. Clinical studies have also reported higher BDNF levels in the PFC of females compared to males [42]. Interestingly, the present study found similar trends in the SNAP25 and VAMP2 protein levels, with females having significantly higher levels of each in the PFC compared with males. In the hippocampus, lower levels of mBDNF were found in females, with a trend indicating a further reduction in stressed females. Müller et al. [40] reported no changes to hippocampal SNAP-25, and RS, in our study, similarly had no effects on the SNAP-25 levels in males. However, the hippocampal SNAP25 levels were significantly reduced in stressed females; thus, hippocampal SNAP-25 may have been impaired by chronic stress in a sex-dependent manner. Previous studies of chronic RS reported reduced hippocampal PSD95 in male mice [24,43], although their protocols did not involve the randomisation of the duration of restraint, as in the present study. Thus, the increase in hippocampal PSD95 may have been a result of a compensatory mechanism due to the possible reduction of presynaptic neurotransmitter release; however, further studies are needed to investigate these mechanisms.

Previously, it was shown that acute RS had no effect on the TH protein levels in the adrenal medulla [44,45]. Furthermore, repeated RS was shown to increase both the protein levels as well as the activity of TH in the adrenal glands [45]. Xu et al. [45] used male Sprague–Dawley rats in their study, subjecting them to repeated immobilization events of 2 h/day for 1 week. In our study, C57BL6 mice were subjected to randomized repeated immobilization events of 2, 3, 4, 5 or 6 h/day (20 h per week, on average) for 5–6 days per week, for 4 weeks. We found randomized repeated RS resulted in no changes in the TH protein levels in the adrenal medullae of both male and female mice, contrary to the findings of Sabben and Kvetnansky, which showed increased TH protein levels and activity following 1 week of repeated immobilization stress in rats [46]. Our interpretation of this result is that repeated RS after 1 week may have caused increased levels of TH protein in males (based on the rat studies described above), but this increase was not sustained after a longer exposure to RS (4 weeks), and the TH protein levels may have returned to basal levels. Interestingly, the female mice showed increased phosphorylation of TH at both the Ser31 and the Ser40 sites, suggesting increased capacity to synthesise catecholamines. Therefore, our findings indicate that the activation of TH in response to RS is sex-dependent. A study by Stroth and Eiden [47] found a single exposure to RS resulted in increased levels of adrenal PNMT mRNA in a time-dependent manner (restraint for 1 h, restraint for 1 h then release to home cage for 5 h, and restraint continuously for 6 h). It was also shown that repeated RS increased the PNMT mRNA levels, leading to increased PNMT protein levels, but following 7 days of RS, the PNMT mRNA and protein levels did not correspond anymore, suggesting desensitisation [48]. In our study, we found no changes in the PNMT protein levels in both male and female mice subjected to repeated and randomized RS. Thus, RS resulted in no changes in the catecholamine synthetic capacity of the adrenal glands in male mice; however, the synthetic capacity was increased in females due to increased TH activity. Overall, our findings suggest that female mice respond differently from male mice during chronic RS due to the increased ability to synthesise catecholamines by the adrenal gland, which may have contributed to the increased anxiety-like behaviour identified in female, but not male, mice.

### Study Limitations

This study has some limitations as it did not measure the free concentrations of tryptophan and its metabolites as well as of catecholamines to confirm that any alterations in the protein levels of the regulatory enzymes investigated led to alterations in outputs of the pathways investigated.

A further limitation of this study is that susceptibility or resilience to stress was not accounted for. For example, a previous study found that both juvenile and adult mice exposed to a 14-day chronic restraint stress protocol exhibited stress resilience, possibly due to habituation to the stressor, with limited changes in behaviour in tests such as the SPT and the EPM, especially in adult mice of a similar age to that of those used in our study [49]. Studies of other models of chronic stress, such as chronic mild unpredictable stress, also implicated adaptations to stress within the hippocampus, finding animals which are resilient to chronic stress had preserved neuronal activation [50], while animals which had enhanced hippocampal BDNF demonstrated stress-resilient behaviours in the SPT [51]. Furthermore, as discussed above, the particular inbred strain of rodent employed in studies of chronic stress may influence their susceptibility to stress depending on which neural circuits are engaged for stress-associated memory consolidation in that particular strain. Thus, the present study’s findings are limited to C57BL6 mice, warranting future studies to explore potential sex differences in different inbred strains in chronic restraint stress.

## 4. Materials and Methods

### 4.1. Animals

All mice were bred in the Core Animal Facility (University of South Australia, Adelaide, Australia), maintained under standard conditions (12:12 h light/dark cycle, lights on between 6 a.m. and 6 p.m., temperature of 22 ± 1 °C, humidity of 52 ± 2%) and housed in groups of 4–5 per cage. All mice were acclimatized to their environment 1 week prior to conducting the experiments and were provided with free access to conventional standard chow and water. On completion of the experiment, all mice were anaesthetized and humanely killed. Brain tissue and adrenal glands were then collected and kept at −80 °C for further analyses. All animal procedures were in compliance with the protocols approved by the Animal Ethics Committee of the University of South Australia.

### 4.2. Experimental Design

Fifty-eight 8–12-week-old C57BL6 mice were separated into experimental groups by sex as follows: male controls (n = 14), male restraint stress (n = 14), female controls (n = 15) and female restraint stress (n = 15). Both the male and the female restraint stress groups were taken through a 4-week restraint stress protocol which involved restraining each mouse daily for a period of 0, 2, 3, 4, 5, or 6 h at random (on average 20 h per week) in a 50 cm cylindrical tube with holes to allow ventilation.

### 4.3. Behavioural Tests

Behavioural testing commenced on the day following the final day of the RS protocol. All behavioural tests commenced at 9:00 am on the day of each test. All mice were acclimated to the testing environment for 10 min prior to each behavioural test.

#### 4.3.1. Open Field Test (OFT)

The open field test was performed as described previously [52]. Briefly, each mouse was placed in the same corner of a white plexiglass square apparatus (40 cm length × 40 cm width × 40 cm height) and allowed to freely explore the apparatus for a 5 min duration. Meanwhile, the activity of each mouse was recorded using an overhead camera and ANY-maze video tracking software (ANY-maze version 7.01, Stoelting, Wood Dale, IL, USA).

#### 4.3.2. Elevated Plus-Maze (EPM) Test

The elevated plus-maze comprises, as previously described [52], an elevated (40–70 cm from floor) plus-shaped apparatus with two opposing closed arms with walls measuring 10 cm in height and two opposing open arms with no walls. Each mouse was placed in the centre of the maze and allowed to freely explore the maze for 5 min. during which each animal’s activity was recorded using ANY-maze video tracking software (ANY-maze version 7.01, Stoelting, Wood Dale, IL, USA).

#### 4.3.3. Tail Suspension Test (TST)

The TST was carried out as previously described [52]. Briefly, each mouse was suspended by its tail using a sticky tape, and a camera was placed at a distance, but perpendicular to the body, in full view of the mouse. Each mouse’s response was recorded for a duration of 6 min. Immobility was considered as the mouse remaining completely motionless without struggling. The total immobility time was counted manually using a stopwatch and recorded.

#### 4.3.4. Sucrose Preference Test (SPT)

Anhedonia was measured using the SPT as described previously [52]. All mice were housed individually in testing cages prior to and during the test. Each mouse was provided with two dual-bearing sipper tubes containing a 1.5% sucrose solution for a 24 h period. Following this, one of the sipper tubes was replaced with regular drinking water for a period of 36 h, with the bottles being swapped twice daily to avoid place preference. All mice were subsequently fasted for a 15 h period. All mice were then tested individually and given a choice of either the 1.5% sucrose solution or drinking water in sipper tubes for a 1 h period. The tubes of both the 1.5% sucrose solution and the drinking water were weighed using an electronic scale prior to testing and were weighed again at the completion of the test. The data are expressed as percentage of sucrose intake compared with the total volume of liquid intake (drinking water intake + sucrose solution intake).

### 4.4. Fresh Tissue Collection and Homogenisation

Using a Precellys 24 Homogeniser (Bertin Technologies, Montigny-le-Bretonneux, France), tissues were homogenized in RIPA buffer (50 mM Tris, 150 mM sodium chloride, 1 mM ethylenediaminetetraacetic acid, 0.5% Triton X-100, 0.5% sodium deoxycholate, pH 7.4) plus a cocktail of protease inhibitors (Sigma-Aldrich, St. Louis, MO, USA), whereafter the homogenates were centrifuged at 13,000 RPM for 30 min. The supernatants of the homogenates were collected, and the protein concentration estimated using a bicinchoninic acid protein assay kit (Thermo-scientific, Rockford, IL, USA) according to the manufacturer’s instructions. The remaining supernatant was used to determine the total protein concentration using a micro-BCA commercial kit (Thermo Fisher Scientific, Waltham, MA, USA) and for subsequent western blot analyses. Discrepancies in the sample numbers in the tissues analyses are due to tissue collection challenges and inadequate sample volumes. Tissues were only collected for male controls (n = 6), male RS (n = 7), female controls (n = 7) and female RS (n = 7).

### 4.5. Western Blot

In total, 30 μg of protein for each hippocampal, prefrontal cortex, colon and adrenal sample was separated by gel electrophoresis on 12% SDS-polyacrylamide gels using the CBS gel system (C.B.S Scientific, San Diego, CA, USA) for 90 min at 120 V. The separated proteins were transferred overnight (960 min) at 100 mA onto 0.2 or 0.45 μm nitrocellulose membranes (GE Healthcare Australia Pty Ltd., Sydney, New South Wales, Australia) and air-dried for 45 min. The air-dried membranes were blocked in 5% skim milk/TBST + 0.05% azide (Sigma-Aldrich) for 1 h at room temperature with gentle shaking, followed washing in TBST. The membranes were incubated overnight at 4 °C with primary antibodies (See Appendix A, Table A1, for details). Following an overnight incubation with the primary antibodies, the membranes were washed in TBST and incubated for 1 h at room temperature with the corresponding secondary antibodies for near-infrared Western blot detection (Li-Cor Biosciences, Lincoln, NE, USA). The resultant immunoblots were imaged using the Odyssey CLx imaging system (LI-COR Biosciences, Lincoln, NE, USA) and quantified using Image Studio Lite 5.2 (LI-COR Biosciences). The proteins were normalised with anti-β-actin.

### 4.6. Statistical Analyses

Two-way ANOVA analyses were carried out using PRISM v8.3 (GraphPad Software, Inc., San Diego, CA, USA) with treatment and sex as between-subject factors. The Shapiro–Wilk normality test was used to determine the distribution of the data prior to the analysis. Post hoc multiple comparisons were made with Bonferroni adjustments. Statistical significance was set at *p* < 0.05.

## 5. Conclusions

The present study identified sex-specific differences in anxiety-like behaviour, synaptic proteins related to the SNARE complex, and adrenal catecholamine synthesis in response to 4 weeks of randomised chronic RS, suggesting female mice may be more vulnerable to an anxiety-like phenotype in this model. Sex differences in basal tryptophan and serotonin regulation were also identified. Future studies in animal models should consider these sex differences, along with the genetic background of the animals used, when evaluating circuits and mechanisms related to chronic stress and depression.

## Figures and Tables

**Figure 1 ijms-24-10353-f001:**
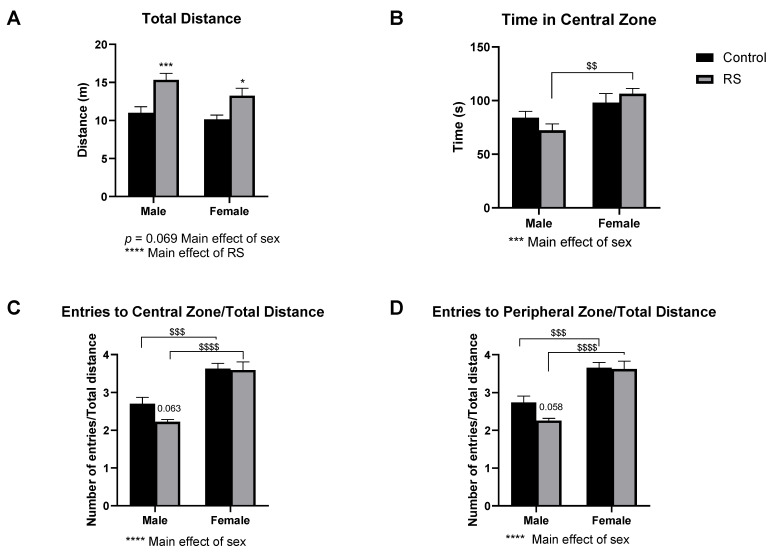
The effects of chronic restraint stress on anxiety-like behaviour and locomotion in the open field test. Measures of total distance (**A**), time spent in the central zone (**B**), number of entries to the central zone corrected per total distance (**C**) and number of entries to the peripheral zone corrected per total distance (**D**) are shown. Two-way ANOVA with Bonferroni multiple comparisons. Male control *n* = 14, male RS *n* = 14, female control *n* = 15, female RS *n* = 13. RS, chronic restraint stress. * (On graph) denotes a significant difference between control and RS means: *p* = 0.063, *p* = 0.058, * = *p* < 0.05, *** = *p* < 0.001. **** = *p* < 0.0001. * (Under graph) denotes a difference in main effects. $ Denotes a difference between male and female groups: $$ = *p* < 0.01, $$$ = *p* < 0.001, $$$$ = *p* < 0.0001. RS, chronic restraint stress.

**Figure 2 ijms-24-10353-f002:**
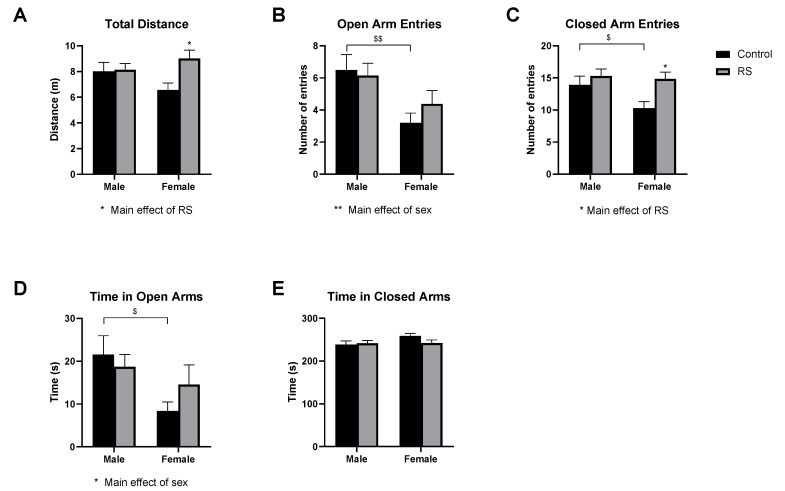
The effects of chronic restraint stress on anxiety-like behaviour and locomotion in the elevated plus-maze test. Measures of total distance (**A**), number of open arm entries (**B**), number of closed arm entries (**C**), time spent in the open arms (**D**), and time spent in the closed arms (**E**) are shown. Two-way ANOVA with Bonferroni multiple comparisons. Male control *n* = 14, male RS *n* = 14, female control *n* = 15, female RS *n* = 13. RS, chronic restraint stress. * (On graph) denotes a significant difference between control and RS means: * = *p* < 0.05. * (Under graph) denotes a difference in main effects: * = *p* < 0.05, ** = *p* < 0.01. $ Denotes a difference between male and female groups: $ = *p* < 0.05, $$ = *p* < 0.01. RS, chronic restraint stress.

**Figure 3 ijms-24-10353-f003:**
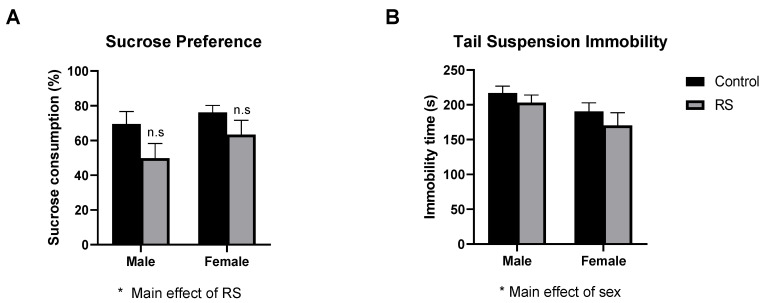
The effects of chronic restraint stress on depression-like behaviour in the sucrose preference test and the tail suspension test. Percentage of sucrose consumption relative to water consumption is shown for the sucrose preference test (**A**) and time spent immobile is shown for the tail suspension test (**B**). Two-way ANOVA with Bonferroni multiple comparisons. SPT: Male control n = 9, male RS *n* = 14, female control *n* = 10, female RS *n* = 15. TST: Male control *n* = 14, male RS *n* = 14, female control *n* = 15, female RS *n* = 12. RS, chronic restraint stress; * (Under graph) denotes a difference in main effects: * = *p* < 0.05. n.s denotes post hoc tests multiple comparisons were not significant. SPT, sucrose preference test; TST, tail suspension test; RS, chronic restraint stress.

**Figure 4 ijms-24-10353-f004:**
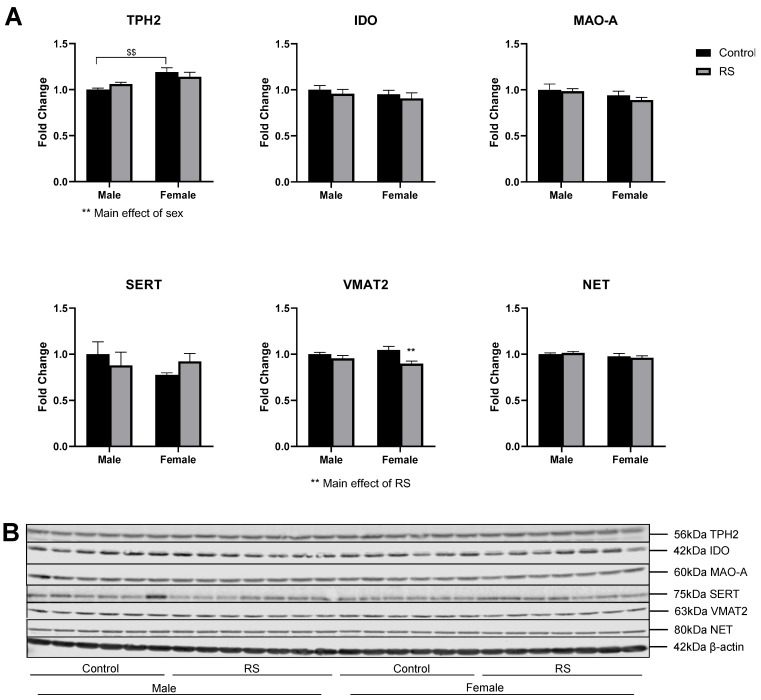
The effects of chronic restraint stress on the regulation of serotonin and noradrenaline pathways and transporters in the prefrontal cortex. (**A**) Two-way ANOVA analyses with all groups quantified relative to male controls with Bonferroni multiple comparisons. Male control *n* = 6, male RS *n* = 7, female control *n* = 6, female RS *n* = 7. ** (On graph) denotes a significant difference between control and RS means: ** = *p* < 0.01. ** (Under graph) denotes a difference in main effects: ** = *p* < 0.01. $ Denotes a difference between male and female groups: $$ = *p* < 0.01. (**B**) Representative immunoblots for TPH2, IDO, MAO-A, SERT, VMAT2, NET and β-actin. TPH2, tryptophan hydroxylase 2; IDO, indoleamine-pyrrole 2,3-dioxygenase; MAO-A, mono-amine oxidase A; SERT, serotonin transporter; VMAT2, vesicular monoamine transporter 2; NET, noradrenaline transporter RS, chronic restraint stress.

**Figure 5 ijms-24-10353-f005:**
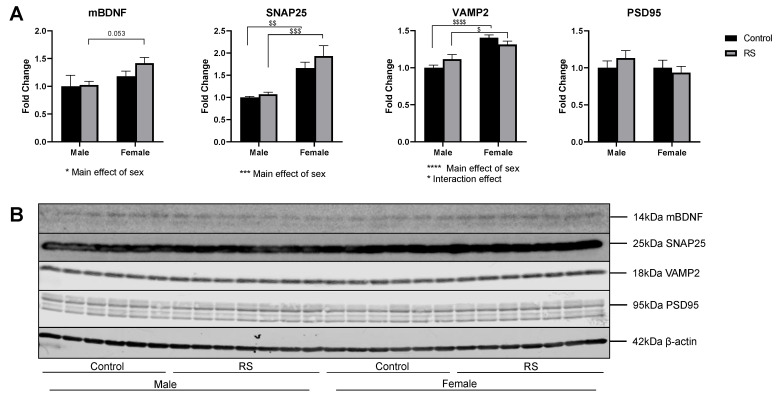
The effects of chronic restraint stress on mBDNF and the synaptic protein levels of SNAP25, VAMP2 and PSD95 in the prefrontal cortex. (**A**) Two-way ANOVA analyses with all groups quantified relative to male controls with Bonferroni multiple comparisons. Male control *n* = 6, male RS *n* = 7, female control *n* = 6, female RS *n* = 7. * (Under graph) denotes a difference in main effects: * = *p* < 0.05, *** = *p* < 0.001, **** = *p* < 0.0001. $ Denotes a difference between male and female groups: *p* = 0.053, $ = *p* < 0.05, $$ = *p* < 0.01, $$$ = *p* < 0.001, $$$$ = *p* < 0.0001. (**B**) Representative immunoblots for mBDNF, SNAP25, VAMP2, PSD95 and β-actin. mBDNF, mature brain-derived neurotrophic factor; SNAP25, synaptosome-associated protein 25; VAMP2, vesicle-associated membrane protein 2; PSD95, postsynaptic density protein 95; RS, chronic restraint stress.

**Figure 6 ijms-24-10353-f006:**
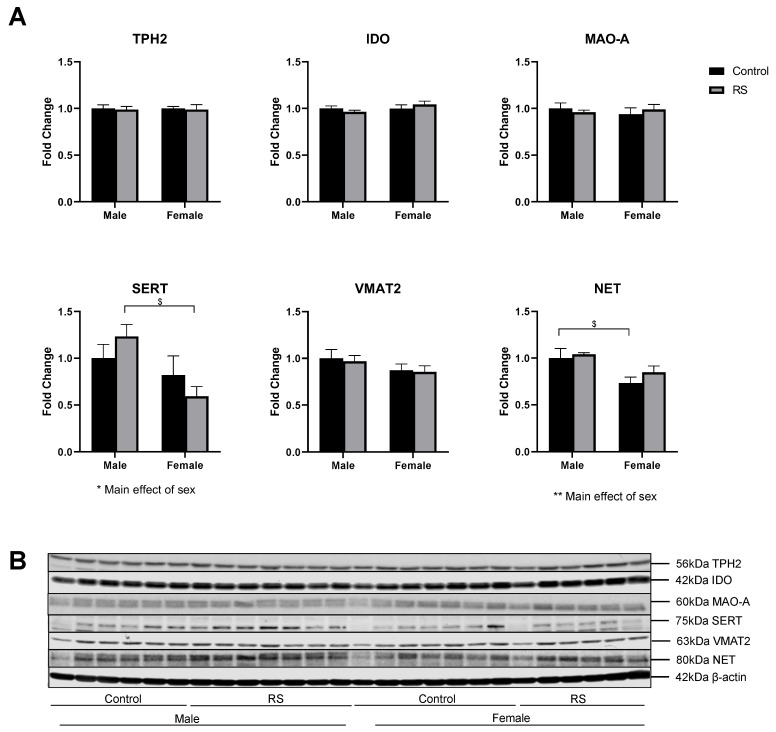
The effects of chronic restraint stress on the regulation of serotonin and noradrenaline pathways and transporters in the hippocampus. (**A**) Two-way ANOVA analyses with all groups quantified relative to male controls with Bonferroni multiple comparisons. Male control *n* = 6, male RS *n* = 7, female control *n* = 7, female RS *n* = 6. * (Under graph) denotes a difference in main effects: * = *p* < 0.05, ** = *p* < 0.01. $ Denotes a difference between male and female groups: $ = *p* < 0.05. (**B**) Representative immunoblots for TPH2, IDO, MAO-A, SERT, VMAT2, NET and β-actin. TPH2, tryptophan hydroxylase 2; IDO, indoleamine-pyrrole 2,3-dioxygenase; MAO-A, mono-amine oxidase A; SERT, serotonin transporter; VMAT2, vesicular monoamine transporter 2; NET, noradrenaline transporter RS, chronic restraint stress.

**Figure 7 ijms-24-10353-f007:**
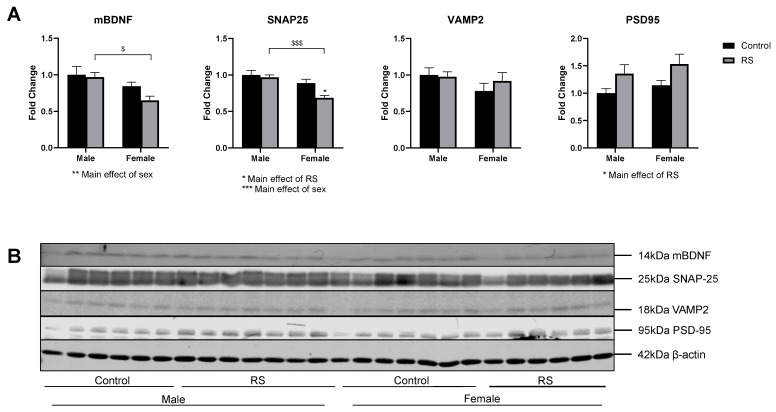
The effects of chronic restraint stress on mBDNF and the synaptic protein levels of SNAP25, VAMP2 and PSD95 in the hippocampus. (**A**) Two-way ANOVA analyses with all groups quantified relative to male controls with Bonferroni multiple comparisons. Male control *n* = 6, male RS *n* = 7, female control *n* = 7, female RS *n* = 6. * (On graph) denotes a significant difference between control and RS means: ** = *p* < 0.01. * (Under graph) denotes a difference in main effects: * = *p* < 0.05, ** = *p* < 0.01, *** = *p* < 0.001. $ Denotes a difference between male and female groups: $ = *p* < 0.05, $$$ = *p* < 0.001. (**B**) Representative immunoblots for mBDNF, SNAP25, VAMP2, PSD95 and β-actin. mBDNF, mature brain-derived neurotrophic factor; SNAP25, synaptosome-associated protein 25; VAMP2, vesicle-associated membrane protein 2; PSD95, postsynaptic density protein 95; RS, chronic restraint stress.

**Figure 8 ijms-24-10353-f008:**
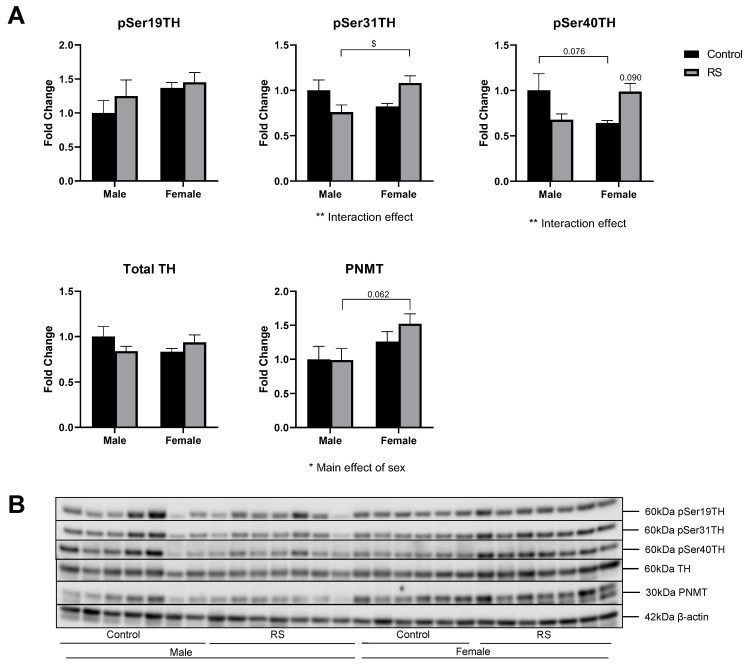
The effects of chronic restraint stress on the regulation of catecholamine synthesis in the adrenal glands. (**A**) Two-way ANOVA analyses with all groups quantified relative to male controls with Bonferroni multiple comparisons. Male control *n* = 7, male RS *n* = 7, female control *n* = 6, female RS *n* = 7. * (Under graph) denotes a difference in main effects: *p* = 0.090, * = *p* < 0.05, ** = *p* < 0.01. $ Denotes a difference between male and female groups: *p* = 0.076, *p* = 0.062, $ = *p* < 0.05. (**B**) Representative immunoblots for pSer19TH, pSer31TH, pSer40TH, Total TH, PNMT and β-actin. pSer19TH, pSer31TH, pSer40TH: phosphorylation residues 19, 31 and 40 of tyrosine hydroxylase; TH, tyrosine hydroxylase; PNMT, phenylethanolamine N-methyltransferase; RS, chronic restraint stress.

## Data Availability

The data that support the findings of this study are available from the corresponding author upon reasonable request.

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
