# Peer review of "Sex-Dependent Effects of Chronic Restraint Stress on Mood-Related Behaviours and Neurochemistry in Mice"

_ijms, 2023, doi:10.3390/ijms241210353_

Round 1

Reviewer 1 Report

Herselman and colleagues report sex-dependent effects of chronic restraint stress on mood behaviours and neurotransmitter pathways in mice. Overall, this is an interesting study mostly because the Authors considered the inclusion of sex a key biological factor. Despite these findings would be of general interest to this field of research, the work raises concerns that need to be addressed.

Major points

-       The interpretation of the results obtained and the conclusions made by the Authors are questionable. The Authors wrote in the abstract:Stressed female mice showed greater anxiety-like behavior than male counterparts”. This statement arises by the fact that female RS mice performed more closed arm entries than control female mice. However, anxiety-like behavior is basically measured by measuring the number of entries and the time spent in the OPEN ARMS. In this study there are only basal differences for these parameters. Moreover, the interpretation made by the Authors is not supported by the results concerning the time spent in the closed arms (no differences). These results instead suggest that RS increased locomotor activity of female mice in the EPM also because paradoxically there was an increase in the number of entries in the open arms.

-       Regarding the results of the sucrose preference test, it is not clear if the post hoc multiple comparison revealed significant differences.

-       A limitation of this study is that the Authors did not take into account stress susceptibility/resilience. There are studies showing stress susceptibility/resilience of rodents to restraint stress (both acute PMID: 33392367, and chonic PMID: 21430148). The Authors should at least discuss this.

-       The title should be adjusted. It would better mood-related behaviors. What does it mean “neurotransmitter pathways?

Minor points

-       Some statements should be rewritten: a near significant effect…

-       There are no letters indicating the graphs in the figures 1 and 2.

-       The f values of the ANOVA should be included.

Some statements should be rewritten.

Author Response

Reviewer 1

The authors would like to thank Reviewer 1 for their thorough review of our manuscript. Please see our responses to each comment below.

Herselman and colleagues report sex-dependent effects of chronic restraint stress on mood behaviours and neurotransmitter pathways in mice. Overall, this is an interesting study mostly because the Authors considered the inclusion of sex a key biological factor. Despite these findings would be of general interest to this field of research, the work raises concerns that need to be addressed.

Major points

-       The interpretation of the results obtained and the conclusions made by the Authors are questionable. The Authors wrote in the abstract: “Stressed female mice showed greater anxiety-like behavior than male counterparts”. This statement arises by the fact that female RS mice performed more closed arm entries than control female mice. However, anxiety-like behavior is basically measured by measuring the number of entries and the time spent in the OPEN ARMS. In this study there are only basal differences for these parameters. Moreover, the interpretation made by the Authors is not supported by the results concerning the time spent in the closed arms (no differences). These results instead suggest that RS increased locomotor activity of female mice in the EPM also because paradoxically there was an increase in the number of entries in the open arms.

The authors acknowledge the criticism of how our interpretation of these results may have been misleading. We, the authors, agree with this criticism and the manuscript has been modified with the interpretation that RS increased locomotor activity of female mice in the EPM.

-       Regarding the results of the sucrose preference test, it is not clear if the post hoc multiple comparison revealed significant differences.

Figure 3 has now been annotated clearly to show that post hoc multiple comparisons revealed no significant differences.

-       A limitation of this study is that the Authors did not take into account stress susceptibility/resilience. There are studies showing stress susceptibility/resilience of rodents to restraint stress (both acute PMID: 33392367, and chonic PMID: 21430148). The Authors should at least discuss this.

The authors would like to thank Reviewer 1 for raising this important limitation. Upon review of the literature this limitation of the study is now included in the discussion in Section 3.1 (Study limitations).

-       The title should be adjusted. It would better mood-related behaviors. What does it mean “neurotransmitter pathways?

The title has been adjusted and made clearer to readers.

Minor points

-       Some statements should be rewritten: a near significant effect…

Statements regarding near significant effects have been removed or rewritten as appropriate.

-       There are no letters indicating the graphs in the figures 1 and 2.

 Lettering has been included in Figures 1 and 2 and in the corresponding figure captions.

-       The f values of the ANOVA should be included.

The f values for all ANOVA statistics have now been included.

Reviewer 2 Report

The study by Herselman et al. is interesting and well done, and the data support the conclusions. I have only minor requests and bibliographic suggestions to relate the results to other studies.

The first sentence of the discussion, "Behavioural testing using the OFT showed that four weeks of chronic restraint stress did not affect thigmotactic behavior in either sex, although males spent less time in the central zone overall compared with females." is not consistent: Spending less time in the core seems to me to affect thigmotaxis behavior.

Furthermore, one of the study's limitations is that it is based on a single crossed strain, when there is an ample demonstration of effects influenced by the genetic background obtained by comparing different inbred strains precisely in the brain structures investigated by the authors and with similar behavioral paradigms—for example, PMID 24667495 and 18184321. Authors should consider this literature when describing their study's limitations and the generalizability of their results.

Author Response

The authors would like to thank Reviewer 2 for their thorough review of our manuscript. Please see our responses to each comment below.

The study by Herselman et al. is interesting and well done, and the data support the conclusions. I have only minor requests and bibliographic suggestions to relate the results to other studies.

The first sentence of the discussion, "Behavioural testing using the OFT showed that four weeks of chronic restraint stress did not affect thigmotactic behavior in either sex, although males spent less time in the central zone overall compared with females." is not consistent: Spending less time in the core seems to me to affect thigmotaxis behavior.

The authors acknowledge this discrepancy in the discussion. This has been modified to convey a clear and consistent interpretation of our reported results.

Furthermore, one of the study's limitations is that it is based on a single crossed strain, when there is an ample demonstration of effects influenced by the genetic background obtained by comparing different inbred strains precisely in the brain structures investigated by the authors and with similar behavioral paradigms—for example, PMID 24667495 and 18184321. Authors should consider this literature when describing their study's limitations and the generalizability of their results.

The authors would like to thank Reviewer 2 for highlighting this important limitation. Upon review of the literature, the authors have included consideration of the influence of the inbred strain used in the study’s limitations and this has also been considered in the discussion of the generalizability of the results.

Reviewer 3 Report

The manuscript presents the results of a comparison of male and female mice responses to restraint stress, based on behavioral tests, the levels of tryptophan metabolism and synaptic proteins in the prefrontal cortex and hippocampus, and adrenal catecholamine regulation. The collected data demonstrate that the tryptophan metabolism was not affected by the restraint stress. Interestingly, the levels of synaptic proteins were reduced in the hippocampus in stressed females, and increased in the prefrontal cortex of all female mice while no such changes were noted in males. Moreover, stressed females but not males showed increased capacity for catecholamine biosynthesis.

The experiments were properly done, described and discussed.

Remarks:

The inclusion of a group of ovariectomized females would allow for evaluation of the effect of sex hormones; this remark is not meant as a critical one but only a suggestion to be taken into account in further studies.

Lines 102 and 104: what are the units of the distance traveled

Lines 109 and 110: what are the units of time?

The units are given in the Figure but should be also in the text

Lines 494-495: no reason to write some reagents starting from capital letters; if any, Tris could be so written

Author Response

The authors would like to thank Reviewer 3 for their thorough review of our manuscript. Please see our responses to each comment below.

The manuscript presents the results of a comparison of male and female mice responses to restraint stress, based on behavioral tests, the levels of tryptophan metabolism and synaptic proteins in the prefrontal cortex and hippocampus, and adrenal catecholamine regulation. The collected data demonstrate that the tryptophan metabolism was not affected by the restraint stress. Interestingly, the levels of synaptic proteins were reduced in the hippocampus in stressed females, and increased in the prefrontal cortex of all female mice while no such changes were noted in males. Moreover, stressed females but not males showed increased capacity for catecholamine biosynthesis.

The experiments were properly done, described and discussed.

Remarks:

The inclusion of a group of ovariectomized females would allow for evaluation of the effect of sex hormones; this remark is not meant as a critical one but only a suggestion to be taken into account in further studies.

The authors agree that this is an excellent suggestion for the evaluation of the effect of sex hormones and this will be taken into account for future studies.

Lines 102 and 104: what are the units of the distance travelled

The units of the distance travelled (metres denoted by “m”) have been included throughout the results section where necessary.

Lines 109 and 110: what are the units of time?

The units are given in the Figure but should be also in the text

The units of the time (metres denoted by “m”) have been included in the text throughout the results section where necessary.

Lines 494-495: no reason to write some reagents starting from capital letters; if any, Tris could be so written

The capitalisation of reagent names has been amended and only Tris and Triton-X are now capitalised.

Round 2

Reviewer 1 Report

The Authors addressed all the points I raised.

Minor editing.